# Three-Dimensional Encoding Approach for Wearable Tactile Communication Devices

**DOI:** 10.3390/s22249568

**Published:** 2022-12-07

**Authors:** Yan-Ni Lin, Yan-Cheng Li, Song Ge, Jing-Jing Xu, Lian-Lin Li, Sheng-Yong Xu

**Affiliations:** 1Key Laboratory for the Physics & Chemistry of Nanodevices, School of Electronics, Peking University, Beijing 100871, China; 2School of Electronics Engineering and Computer Science, Peking University, Beijing 100871, China; 3School of Microelectronics, Shandong University, Jinan 250100, China

**Keywords:** tactile sensory, pattern encoding, temporal sequence, silent communication, visually impaired users

## Abstract

In this work, we presented a novel encoding method for tactile communication. This approach was based on several tactile sensory characteristics of human skin at different body parts, such as the head and neck, where location coordinates in the three-dimensional (3D) space were clearly mapped in the brain cortex, and gentle stimulations of vibrational touching with varied strengths were received instantly and precisely. For certain applications, such as playing cards or navigating walk paths for blinded people, we demonstrated specifically designed code lists with different patterns of tactile points in varied temporal sequences. By optimizing these codes, we achieved excellent efficiency and accuracy in our test experiments. As this method matched well with the natural habits of tactile sensory, it was easy to learn in a short training period. The results of the present work have offered a silent, efficient and accurate communication solution for visually impaired people or other users.

## 1. Introduction

A human brain works in a dark, closed cranial cavity. The brain receives external information data through body sensors in eyes, ears, nose, tongue, skin, and so on. It analyzes the received data, compares with memorized data, makes decisions and then gives orders for body actions.

Statistical analysis shows that nearly 90% of the external information a brain receives is obtained via visual pictures [1], and sounds such as languages carry the second largest volume of external information. For the unfortunate group of people who have lost eyesight, languages are the most efficient media for communication in daily life as well as in training and learning processes of education.

Tactile sensation is also a strong sensory capability of a human body. Compared to olfactory sensation and gustatory sensation, so-called smelling and tasting in our daily lives, touching sense (tactile sensation) has its unique advantage in that the location coordinates of any part of a human skin in the three-dimensional (3D) space are clearly mapped in certain regions of the brain cortex [2], and gentle touching stimulations can be received instantly and precisely. In other words, the tactile sensation in the human neural system seems having a natural navigation mechanism for the body skins. Moreover, as compared to the language voices, tactile stimuli can be performed in silence as a secret communication method.

Different parts of a human body, such as the arm [3], palm [4,5], head [6], waist [7], tongue surface [8], body back [9,10], finger [11], etc., have been investigated for touching stimulation experiments and tactile communication. Most groups focused on investigating the capacity of tactile coding for delivering text information such as Morse codes. For instance, glove-like devices which consisted of many vibrotactile motors were demonstrated to convey information using the Malossi or the Lorm alphabets [12,13]. Gaffary presented three tactile Roman alphabets based on different temporal, symbolic and spatial communication strategies [11]. Rantala et al. presented devices capable of creating three tactile vibration modes, scan, sweep, and rhythm, for delivering text information [14]. Liu et al. systematically investigated the efficiency of two different sets of Vibraille codes in terms of time and frequency pattern for delivering words and symbols [15].

With the technological developments in fields of brain–computer interface, artificial intelligence and “cyborgs”, tactile communication for information other than alphabets was recently paid more attention. Hao et al. investigated triangular codes with touching devices mounted at the waist position of people under test [16]. Experiments using tactile codes for offering pilots the navigation information or environmental information were demonstrated [17,18].

Yet the potential of tactile encoding has not been fully explored. Since tactile sensation is directly connected to the hunting and escaping instincts in evolution, body reactions for touching stimuli are straightforward and fast. On the other hand, the coordinates of skin at different parts of a human body naturally form different 3D coordinate systems. For example, the shape of head skin resembles a spherical coordinate system used in mathematics. From the heart to the skin, spots on the left hand and right hand, left foot and right root, chest and back, etc., form natural 3D vectors and define unambiguous left, right, front and rear directions in pedestrian navigating. It is feasible to create hybrid encoding rules and codebooks using tactile elements of vectors, patterns, temporal sequences and so on for different practical applications.

In this work, we used the natural 3D coordinates of skin sensory at body parts such as head and neck, optimized several sets of touching codes with suitable 3D patterns and temporal sequences, and demonstrated guiding blinded persons to play card games, navigating them to walk alone in a street and to avoid obstacles in the paths as well as delivering text messages of numbers and alphabets. Because this 3D-pattern-based method matched well with the instinct tactile reactions of the human body, the minimum training process needed for these tactile codes was only tens of minutes.

## 2. Experimental Details

In this work, we developed different sets of wearable devices and related communication systems for tactile communication tests with college students and blind people. In the following, we briefly described our devices and experimental setups.

Body parts such as the top of the head, forehead, temple, neck, wrist, etc., were chosen to test the skin sensitivity in terms of sensitivity to the touching strength, spatial resolution, temporal resolution, fatigue and so on, for our different homemade wearable tactile devices.

After a large number of early experiments, coin motors were chosen as actuators to deliver touching stimulation to the target skin regions. Coin motors with a diameter of 10 mm and thickness of 2.7mm (CY-1027-00-10) were used in our wearable devices. These motors were driven by a DC voltage of 0∼3.3 V. They delivered vibrational stimulation to the skin when they were closely mounted inside a soft-material cover, which was elastic fabric in our experiments, as shown in Figure 1.

The skin sensitivity to the vibrational intensity and duration of the coin motors was tested for different kind of devices fixed as the head, forehead, neck, wrist, waist, arm, etc. Figure 1A showed a typical device used at the neck region, and Figure 1B showed a device for the top of the head. In Figure 1A, the brown, ring-like neck device was embedded with 4 coin motors at the front (F), left (L), back (B) and right (R) position of the neck, respectively. In Figure 1B, 11 coin motors were embedded in the device, eight of which were mounted around the head at the front (F), left (L), back (B), right (R), front right (FR), back right (BR), front left (FL) and back left (BL) positions. For the remaining three, one is mounted at the center of the head top, and the other two were located at halfway to FR and FL, as shown in Figure 1B. Figure 1C showed a photograph of a tester wearing the 11-motor head device, where the power circuit and wireless communication package were mounted on her back. Details of these driving circuits and communication devices were published elsewhere [19]. For head devices, several versions with nine motor and 11 motors were fabricated and tested.

In the neck ring experiment, the coin motors were controlled by a microprocessor on a demo-board Arduino Uno, to which instructions of vibration were sent from a desktop computer by the experimenter. The vibration amplitude was controlled by changing the input voltage (0∼3.3 V DC) of the motor using pulse width modulation (PWM) on an 8-bit (256-step) scale. While a subject sat with his back to the screen answering the queries orally, the experimenter used the keyboard of the computer to send instructions and record subjects’ responses. It was ensured that subjects could not distinguish the instructions by listening to the keystrokes.

Adopting the foregoing setting in the neck ring experiment, we used an adaptive 1-up, 1-down paradigm to investigate the neck’s sensitivity to vibration [20]. The results provided the subsequent experiments of thresholds as a reference of parameter setting, with much greater values adopted, to ensure subjects to well perceive vibration.

The vibration stimulation of the head device was controlled by wireless transmission, and the information was obtained through several video cameras mounted around the experimental room. For pedestrian navigating experiments, the video cameras were carried on the users (e.g., a blind person) under test.

For the results reported in this work, 12 participants were tested, of which 11 were healthy college students. During tests, their eyes were blocked with a black eye-mask, which totally shielded their eyesight. The other one was a visually impaired patient, who participated in the test of navigation codes for the head device.

In the next section, some major results were presented.

## 3. Results

Our spatial resolution results show that the spatial resolution in many parts is only 2∼3 cm, so we ensure that the motor spacing is greater than 5cm in the wearable device [19]. Below, we present some important experimental results.

### 3.1. Sensitivity and Spatial Resolution of Body Skins to Tactile Actuators

We have measured the skin sensitivity to the vibrational intensity and duration of the motor actuators for different parts of body skins, including the head, forehead, neck, wrist, waist, arm, etc. We have also tested the temporal resolution and spatial resolution for the stimuli of multiple actuators mounted at these body parts.

The mean intensity threshold among all subjects was 69.85 (≈0.90 V, SD=10.38≈0.13V). The mean duration threshold was 18.5ms (SD=2.1ms). As the results show in Figure 2, the uncertain range in terms of vibration intensity is 0.49∼1.43 V; the uncertain range in terms of duration is 3.6∼27.7 ms. Therefore, if the vibration intensity is greater than 2.5V and the vibration time is greater than 80ms, the subject’s perception can be guaranteed. This will be used as a reference for parameter setting in subsequent experiments.

### 3.2. Temporal Sequences in Tactile Codes

According to the above results, we used a specific time-series vibration stimulus. The stimulus paradigm for us is that two different motors are vibrating successively with a stimulus onset asynchrony (SOA) to form a stroke. The stroke is expected to give the user the top-to-bottom graphical perception shown in Figure 3a. In order to convey richer information, we combine multiple strokes into a sequence to obtain a symbol. For example, Figure 3b is a symbol with a length of two strokes, and we expect the user to produce a graphical perception of the shape J.

### 3.3. Performance in Card Games

In different application scenarios, different encoding methods can be adopted according to the formulated stimulus paradigm. For example, in playing cards, there are four different colors, which can be corresponding to the four quadrants for coding and discrimination, as shown in Figure 4. Taking the spades color coding as an example, the actuators F and R vibrate in turn, and the vibration perception in the first quadrant of the subject can quickly correspond to the spades coding.

For the numerical coding of a poker, we designed a set of poker codes based on the stereoscopic characteristics of neck perception as shown in Figure 5. Figure 5a shows a dial. Figure 5b shows that the stereo perception of the neck corresponds to the clock, where the motor F corresponds to “12” on the dial, the motor R corresponds to the 3 on the dial and so on. One to three are the F zone; when the motor F vibrates one time, it means that the 12 of the dial moves one space to the right, and the corresponding code is the number “1” and so on. Specific instructions are shown in Figure 5d. Taking the five of hearts as an example, the formal instruction FL given above is transmitted first, which is followed by the digital instruction RR.

In order to verify the feasibility of the above coding, a practical poker test was conducted. The test uses the “Blackjack” rule; this experiment only needs to transmit the digital code. Subject 1 simulated the situation of no visual input by using a blindfold. Subject 2 was a normal population. As shown in Figure 6, the two subjects played a “Blackjack” game. The hand of the subject was “9” and “10”, and the code of “BBB” and “L” was transmitted to the subject 1 in the background. Subject 1 obtained vibration stimulus information, which was identified as “9” and “10”, and chose show hand; the final result was that subject 1 won.

In the poker experiment, after simple training, the participants’ perceptual response time to instructions in the experiments was provided in 1s. Participants were 98% or more correct in perception. Subjects can perform normal activities using existing instruction codes in this application scenario.

Experimental results show that the coding based on human stereoscopic perception has the advantages of fast learning, accurate recognition and accurate cognition. After a short learning time, subjects can quickly master the set of codes within five minutes and make accurate recognition and reaction in the actual scene.

If used in a complex poker game, both suits and numbers can be transmitted simultaneously. We also set a feedback device in the wearable device. If the subject has no perception, it can fed back through the key on the device.

### 3.4. Performance in Navigation Scenarios

Simple information can be transmitted through the three-dimensional perception of the skin. As for the navigation instructions that meet the requirements of normal travel in daily life, taking into account the dependence of navigation on orientation, the three-dimensional perception of skin can be used for instruction coding, as shown in Figure 7.

Similarly, in the head tactile device, we also use the same encoding method to encode navigation instructions, as shown in Figure 8.

The experiment involved a 31-year-old man who was completely blind. The participant suffered from congenital cataracts and was blinded by a failed operation at the age of 1 year. The participant wore the prototype system and walked an L-shaped (straight and turn test) route on a real pedestrian street after becoming familiar with the instructions. During the process, attention should be paid to avoid the coming and going pedestrians, parked vehicles, roadside steps and trees, and pay attention to the turning time, given the turn command in time.

The experiment is shown in Figure 9. The researchers can clearly determine the position of the participant through the video and quickly send corresponding commands, such as forward, stop, left turn and so on. The participant successfully avoided pedestrians, parked vehicles, trees, steps, and obstacles on the road. Experiments showed that the blind participant was able to accurately accept tactile instructions and walk according to the instructions in some relatively simple road conditions.

### 3.5. Codebooks for Text Communication

In addition to using tactile stereo perception directly, when we need to transmit complex text information, we can also construct coding through the graphical perception of the skin. We used Roman numerals for graphical processing, and the specific instructions are shown in Table 1. Taking number 2 for example, as its Roman numeral is II, instruction FBFB would be transmitted for the user to perceive.

Extending from Roman numerals, the Latin alphabet can be constructed according to the same coding principles. Based on the neck ring device, we have constructed 26 Latin alphabets on a graphical basis, as shown in Table 2. Taking the letter “A” as an example, the stimulus instruction is FLFR, that is, the graph perceived by the subject is the graph corresponding to “A” in Table 2.

We evaluated the accuracy of all possible symbols of two strokes in length, setting the temporal parameters based on the result of the former step (100ms in the amount of single vibrating time, 150ms in intra-stroke interval, and 300ms in inter-stroke interval). The possible symbols included 144 different ones, which could be abbreviated as “FLFL”, “FLFB”, etc. (see Figure 3b). All the symbols were duplicated once and shuffled to make a sequence of 288 symbols, and the sequence was shared among the subjects. For each trial, the experimenter typed the abbreviation into the desktop computer to send instructions, after which the subject needed to answer the abbreviation or say “don’t know”. There was a one-minute break between every 24 trials.

The interactions between spatial and temporal features of stimuli may result in perceptual illusions, and movement illusion is one of them [21]. Movement illusion is caused by successive onsets of stimuli on the skin. If the actuation spans of two adjacent motors overlap, the user will perceive a single stimulus moving between the two locations rather than both stimuli. Therefore, in the subsequent experiments, we compared the perceptual rate in condition of with illusion and without illusion.

We selected 26 sets of instructions for the following testing. Before the start of this step, there was visual training for the subjects. A sheet with Table 2 printed was given to the subject. The subject was required to memorize the alphabet without receiving any vibrotactile stimulation. To verify if the subject remembered the alphabet, the experimenter would ask the subject which letter an abbreviation was for 52 trials, each letter occurring twice. The following steps would not be taken until the accuracy achieved 95% (less than two wrong answers); otherwise, another session would be conducted after the subject felt confident to answer the queries.

The confusion matrices with and without movement illusion are shown in Figure 10. For the case without movement illusion, letters of “Y”, “Q”, “U”, “G”, “P”, and “R” had relatively low accuracy. The overall mean accuracy in the case without movement illusion was 0.783(SD=0.203), and that in the case with movement illusion was 0.565(SD=0.282). Using a Kolmogorov–Smirnov test, we found that normality for letter accuracy failed in both cases (p<0.001), so we used Mann–Whitney U Test to compare the two sets of data. We found there was a significant statistical difference between the accuracy of two cases (p<0.001).

We had expected a greater recognition rate with the use of movement illusion, which caters to our concept of stroke. Comparing Figure 10a,b, we can find the letters “N”, “F”, “U”, “Q”, and “W” suffered a severe degradation of accuracy. The frustrating results may be enlightened in future research to explore for a more elaborate encoding design and more proper parameter setting.

## 4. Discussions

A human brain is a complicated data processor which has more around two hundred different functional regions in its cortex. The brain is capable of simultaneously receiving input data from multiple body sensor channels. In driving, for example, the driver can look at the road, enjoy a song, talk to his passenger, taste a candy, and simutaneously handle the steering wheel and step on the accelerator or brake pad.

Therefore, receiving external information through tactile sensation is an additional and alternative input channel. Sending directional information to a pilot through tactile coding, as has performed by Yang and Huang et al., is a valuable demonstration [17,18].

In the pedestrian navigating experiments, for example, a fresh user needed only ten minutes of training to understand the tactile codes delivered to the device mounted on his/her head, and then, the user was able to follow the tactile orders and walk alone in a corridor of a building to find a seat in a café, or walk in a street. Similarly, by using vector-like 3D-pattern-based tactile codes, we could help a blinded person catch a cup on the table, turn a piece of block, complete a puzzle game, etc.

Previously, other than the well-studied tactile communication for text messages, pattern-based tactile codes have been investigated, such as experiments on “triangular codes” with devices on the waist [18] and patterns sent with an electrical device on the tongue surface [8]. Our experiments in the present work extended the pattern-based concept, and it showed the potential of developing a new branch of tactile-encoding methods utilizing the 3D patterns, 3D vectors, as well as temporal sequences for many unexplored application fields.

For instance, hybrid tactile codes consisting of 2D patterns, 3D patterns, intensity distributions and temporal sequences, may be applied in studies of brain functions. One recent model suggested that a brain stored its memory data in the form of clusters of strongly connected neurons. Each cluster is similar to a “2D code”, where neurons of the cluster are strongly connected through synapses, and the excitation of one neuron of the cluster will light all neurons of this cluster, projecting a 2D pattern to the analysis layer; thus, the brain retrieves one piece of memorized information. Therefore, the hybrid tactile codes may mimic the patterns of neuron excitation in a memory cluster to verify the validity of the model.

## 5. Conclusions

In short, we investigated an alternative encoding approach for tactile communication by utilizing the natural sensation characteristics of body skins so that the brain could sensitively and precisely identify the 3D coordinates of body skins at various body parts. Using small vibrational devices as the tactile stimulation actuators, we tested the tactile sensitivity, spatial resolution and learnability of body skins at the head, neck and wrist of local college students. We demonstrated that tactile codes can be designed with a large room of flexibility for different body parts and for different users.

By mimicking the concepts of vectors, rectangular and spherical coordinate systems, as well as clock panels, we demonstrated that blinded persons could play card games, walk alone in street, etc., by receiving gentle stimuli of 3D-pattern-based tactile codes. The card information could be delivered instantly and precisely to the blinded person under test via only four vibrational actuators around the neck. Similarly, pedestrian navigation could be efficiently performed with 8–11 vibrational actuators mounted on the head skin. We also performed tactile encoding with the neck devices for delivering numbers and alphabets and analyzed the recognizing accuracy for testers with different training hours. As this method matched well with the natural instincts of a human tactile sensation system, after training for half an hour, fresh learners could complete the card playing and walking alone tasks. However, the training time needed for precise test communication was much longer.

We have shown some versatile and flexible aspects of the tactile encoding approach using the natural 3D skin coordinates. The results may shed light on developing silent, efficient and accurate and easy-learning communication solutions for visually impaired people or other users.

## Figures and Tables

**Figure 1 sensors-22-09568-f001:**
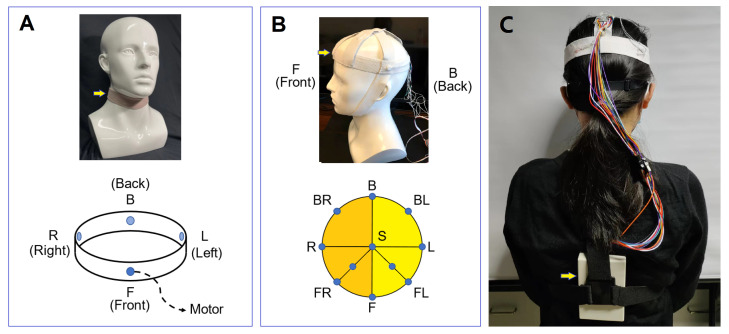
Schematic diagrams of the homemade tactile communication devices tested in this work. (**A**) A brown, ring-like neck device (highlighted by an arrow) was mounted on a gray 1:1 plastic head model. The neck device was embedded with four vibration motor actuators at the front, left, rear and right side of the user’s neck, which were denoted as F, L, B and R respectively. (**B**) A white head device mounted on a 1:1 white plastic head model. This head device was embedded with 11 vibration motor actuators located at the blue dot regions shown in the figure. For instance, the one mounted on the front position of the forehead was labeled as “F”. (**C**) A photograph of a tester wearing the 11-motor head device. The power circuit and wireless communication package were mounted on her back.

**Figure 2 sensors-22-09568-f002:**
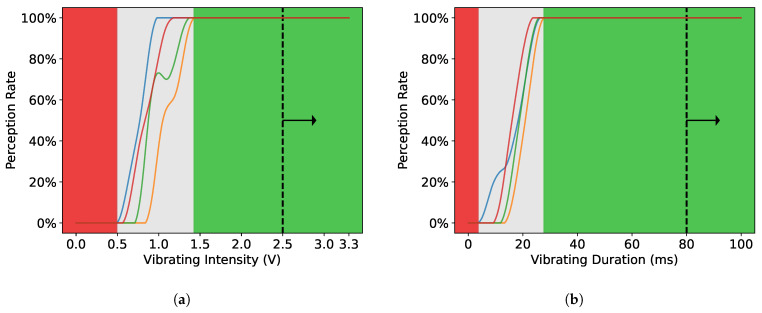
The sensitivity of neck skin to motor actuators in terms of vibrational intensity and duration. The gray region signifies uncertain range, in which the corresponding parameter setting would elicit either positive responses (“yes”) and negative responses (“no”) in the practical trials. All the responses in the red region were negative, while all the responses in the green region were positive. (**a**) Sensitivity in terms of intensity among all subjects. (**b**) Sensitivity in terms of duration among all subjects.

**Figure 3 sensors-22-09568-f003:**
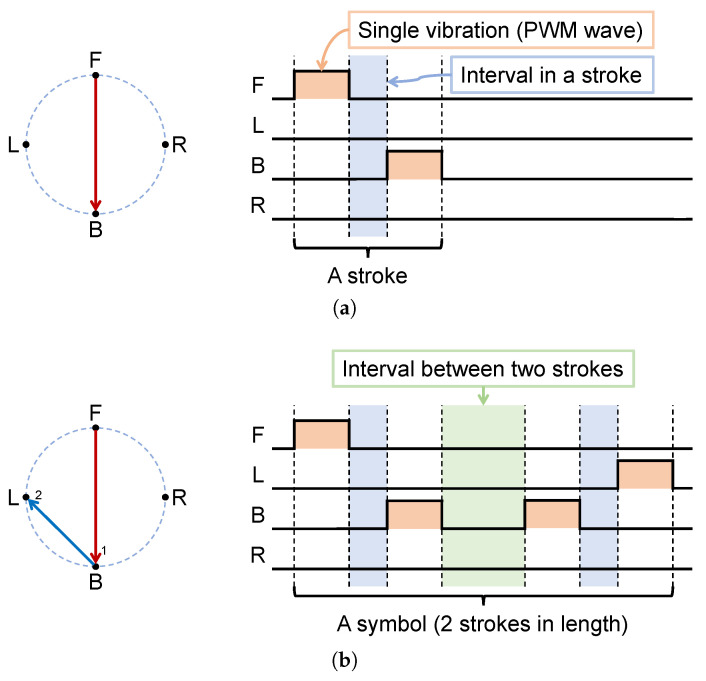
The pattern of motor vibration sequence we used and the corresponding pattern, taking the neck ring as an example. (**a**) Vectorized perception of a pair of stimuli FB and corresponding motor vibration timing diagram. (**b**) Vectorized perception of a complete set of stimuli FBBL and corresponding motor vibration timing diagrams.

**Figure 4 sensors-22-09568-f004:**
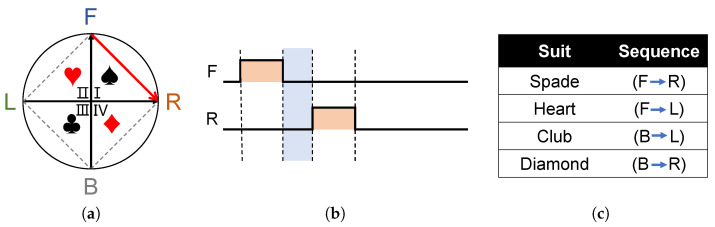
Poker stereoscopic perceptual suit coding, taking spades suit coding FR as an example. (**a**) The neck ring consists of four quadrants with pairwise motors and their corresponding patterns. (**b**) Motor timing diagram corresponding to spades code. (**c**) The poker suit code corresponding to the neck ring.

**Figure 5 sensors-22-09568-f005:**
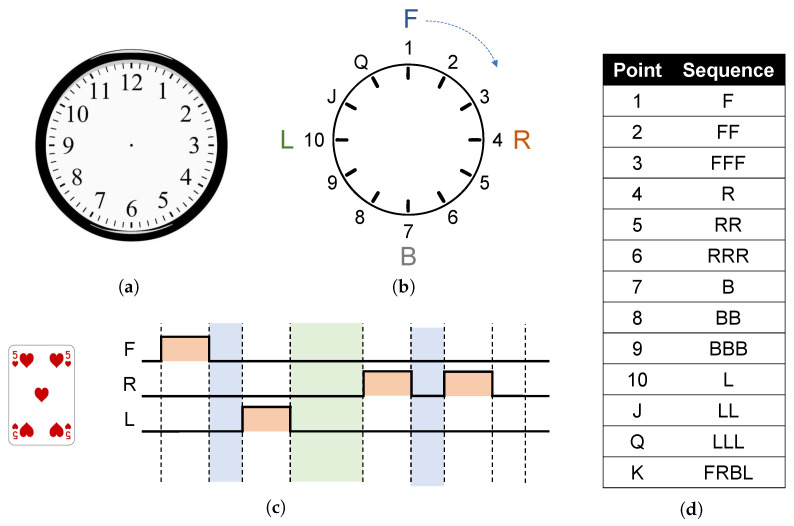
Poker stereoscopic digital coding, taking the 5 of spades coding as an example. (**a**) A dial. (**b**) The three-dimensional distribution of the motor in the neck corresponds to the dial and the number. (**c**) Motor timing diagram corresponding to the 5 of spades code. (**d**) The poker number code corresponding to the neck ring.

**Figure 6 sensors-22-09568-f006:**
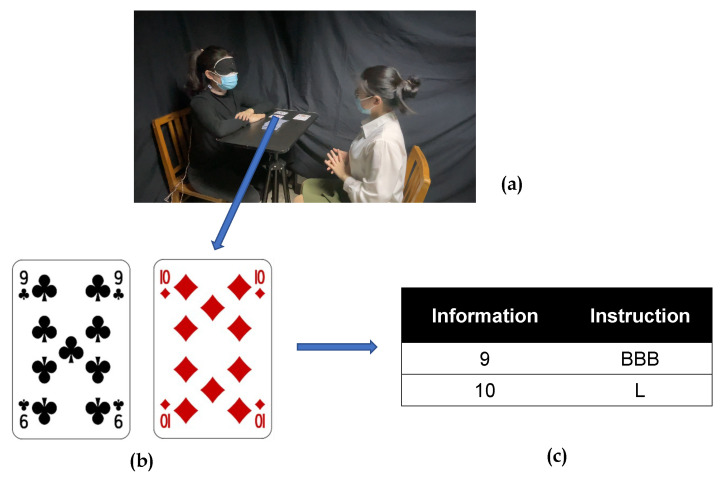
Coding Scenario Application. (**a**) Subjects on the left are patients with visual disabilities, and subjects on the right are normal subjects. Instructions are transmitted through the background. (**b**) The specific card face of the visually impaired subject. (**c**) The corresponding transmission command of the card face.

**Figure 7 sensors-22-09568-f007:**
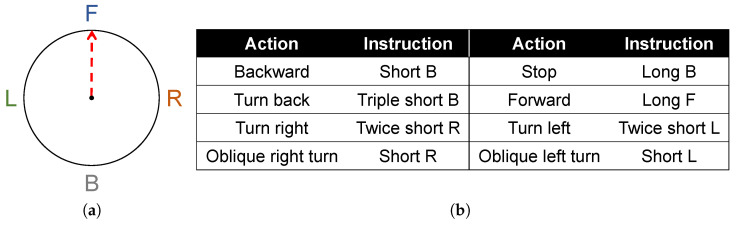
Navigation code of the neck ring device. Take “forward” as an example. (**a**) Stereoscopic perception of forward command on the neck ring. (**b**) Navigation codes corresponding to neck rings.

**Figure 8 sensors-22-09568-f008:**
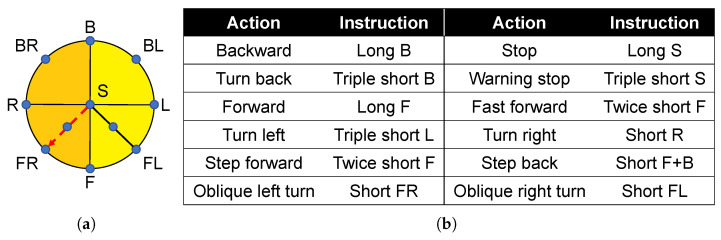
Oblique left turn was used as an example for the navigation coding of the oblique left Turn. (**a**) Stereo perception of the oblique left turn command in the head. (**b**) Navigation codes corresponding to head devices.

**Figure 9 sensors-22-09568-f009:**
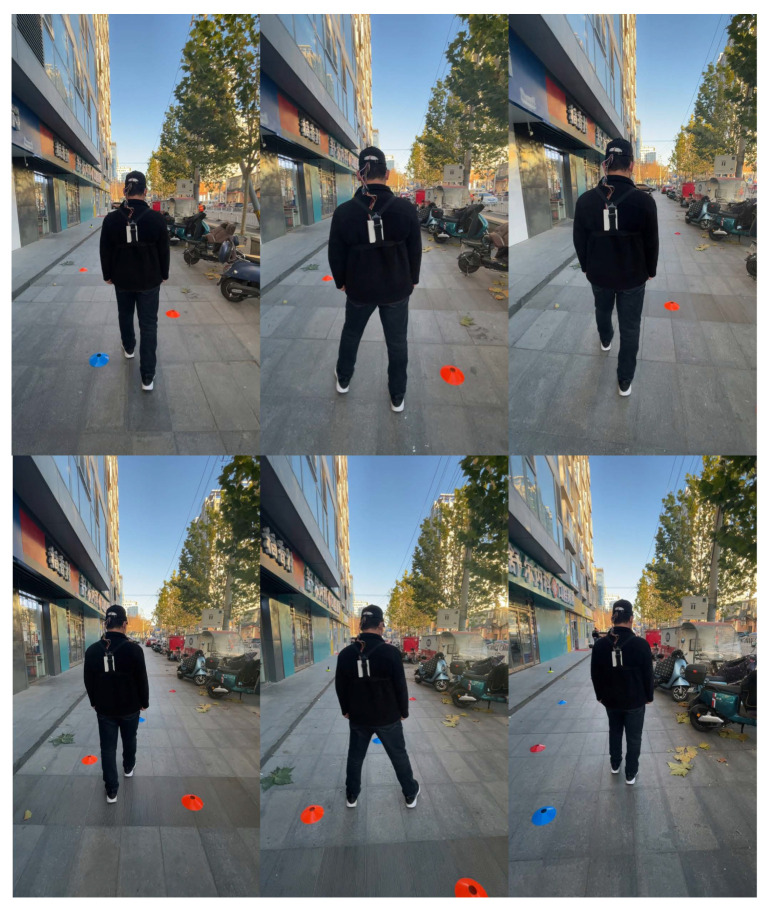
Walking obstacle avoidance experiment of visually impaired patients.

**Figure 10 sensors-22-09568-f010:**
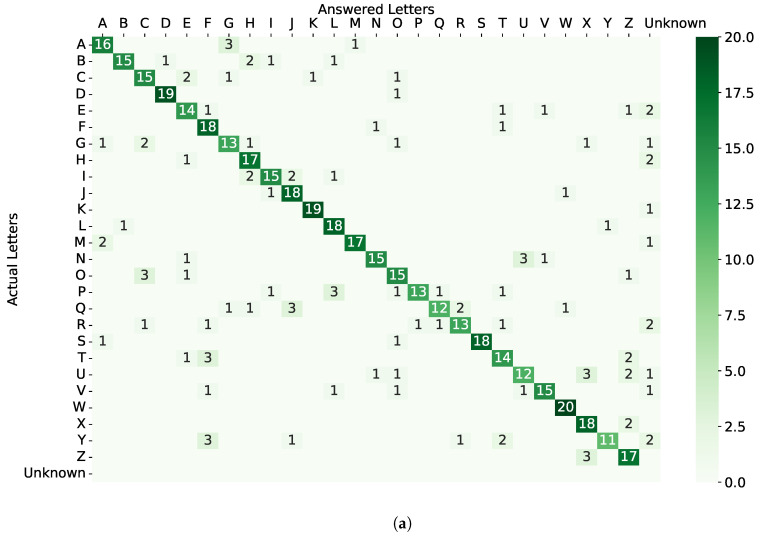
Confusion matrices across all participants. (**a**) Without movement illusion. (**b**) With movement illusion.

**Table 1 sensors-22-09568-t001:** The digital instruction table for the choker. Red arrows in the pattern columns indicate the first group of strokes, while blue arrows indicate the second.

**Numeral**	**Symbol**	**Pattern**	**Numeral**	**Symbol**	**Pattern**	**Numeral**	**Symbol**	**Pattern**
1	FB	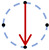	2	FBFB	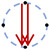	3	FBFBFB	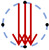
4	FBLBBR	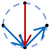	5	LBBR	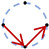	6	LBBRFB	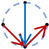
7	LBBRFBFB	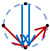	8	LBBRFBFBFB	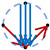	9	FBLRRL	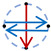
10	LRRL	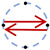						

**Table 2 sensors-22-09568-t002:** The vibrotactile alphabet for the choker. In the columns of symbols, abbreviations of two strokes are given (see Figure 3). Red arrows in the pattern columns indicate the first stroke, while blue arrows indicate the second.

**Letter**	**Symbol**	**Pattern**	**Letter**	**Symbol**	**Pattern**	**Letter**	**Symbol**	**Pattern**
A	FLFR	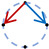	J	FBBL	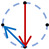	S	FLRB	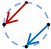
B	FBRB	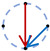	K	LFLB	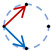	T	LRFB	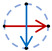
C	FLLB	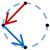	L	FBBR	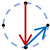	U	LBRF	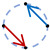
D	FRRB	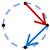	M	LFFR	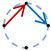	V	LBBR	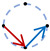
E	LRLB	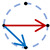	N	LBFR	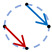	W	RBBL	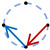
F	FBLR	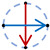	O	FLBR	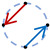	X	LRRL	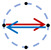
G	FLLR	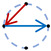	P	FBFR	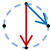	Y	FBLB	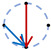
H	RLFB	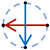	Q	FBFL	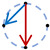	Z	LRRB	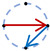
I	FBBF	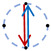	R	FBLF	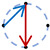			

## Data Availability

Not applicable.

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
