# Peer review of "Three-Dimensional Encoding Approach for Wearable Tactile Communication Devices"

_sensors, 2022, doi:10.3390/s22249568_

Round 1

Reviewer 1 Report

First of all, why the author wanted to test this configuration, The motor placement seems so basic and has been examined before, as cited in the paper, therefore what's the novelty here? A new (arbitrary) coding system? then, why it's compared to previous attempts?

Secondly, this paper is written extremely sloppy and shouldn't be accepted as it is. Following are the points which should be addressed.

# L96, The motor doesn't respond linearly upon the given PWM value. Measure the vibration intensity in the physical unit (such as displacement and acceleration) according to the assigned PWM value. Similarly, an ERM motor doesn't respond to the short pulse of signals. 

# L101-109, the writing style is wholly unacceptable and unscientific. In general, the term "some" SHOULD NOT BE USED STRICTLY. Specify the exact numbers and descriptions, which should be detailed enough when some other is trying to reproduce the experiment presented in the paper.

  •  "more than ten", "Some were" -> specify the exact number.
  •  "Standard testing procedure" -> there's no such thing as "standard" in this kind of experiment. Present a reproducible experiment procedure.
  • "adjustments were made" -> what are they? They should be precisely specified if they're altering the experiment procedure and affecting the result.
  • "Feedbacks in terms of sound, language, and written test sheet" -> Ambiguous. Specify.

# L115, Where are the results of the "spatial resolution"? 

# L119, "69.85" -> what's the unit of this value?

# L121, "range of fuzzy perception" -> How do you define it? It seems arbitrarily set by eyeballing.

# The results presented in Figure 2 are not meaningful because it depends on the used motor's characteristics. Also, there might be an interaction between applied voltage and duration -- which is not specified here. Measure and present the motor's characteristics; for example, at which voltage it starts spinning, how much acceleration it produces on each voltage level, how long the pulse should be applied to make it spin on various voltage levels, etc.

#L131, Present the feedback design for all symbols used for the experiment.

#L158, Specify the experiment procedure, stimuli, responses, analysis, etc. Not just present "98% were correct". Especially for a poker game, transmitting the information of one card is not enough -- it is supposed to be a more complex scheme, which is entirely missing in this paper.

#Figure7 and Figure8. How the authors came up with the mapping between the action and instruction? It looks pretty much arbitrary without supporting backgrounds.

#L176, only one participant is insufficient to prove the system's usability. Especially on Figure 8, the authors introduced 12 actions-instruction mappings, which should be appropriately verified based on the recall rates measured from multiple participants.

L200-207, I can't understand the experiment procedure here at all. Extremely confusing. What's the "abbreviation"? And where's the result of this experiment?

L211, isn't it suppose to be Table 2?

L209, "Movement illusion" is introduced here firstly and became the independent variable of the following experiment, but never explained adequately.

Reviewer 2 Report

1. Explain why the spatial resolution in many parts is only 2 ~ 3 cm ?

2. Please justify "Experimental results show that the coding based on human stereoscopic perception has the advantages of fast learning, accurate recognition and accurate cognition. How and Why?

3. What language has been used for coding?

4. Have you compared these results with any of the earlier published data or with some analytical data. Please comment. 
